# Evaluation of the efficacy of using indocyanine green associated with fluorescence in sentinel lymph node biopsy

**Rafael da Silva Sá**[1,2,3][ID]*, **Raquel Fujinohara Von Ah Rodrigues**[2‡], **Luiz Antônio Bugalho**[2‡], **Suelen Umbelino da Silva**[3‡], **Afonso Celso Pinto Nazário**[1]

**1** Discipline of Mastology, Department of Gynecology, Universidade Federal de São Paulo (UNIFESP), São Paulo, São Paulo, Brazil, **2** Department of Gynecology and Mastology, Hospital de Esperança, Presidente Prudente, São Paulo, Brazil, **3** Universidade do Oeste Paulista (UNOESTE), Presidente Prudente, São Paulo, Brazil

☯ These authors contributed equally to this work.
‡ RFVAR, LAB and SUS also contributed equally to this work.
* rafasamed@hotmail.com

**Data Availability Statement:** All relevant data are within the paper and its Supporting information files.

## Abstract

### Introduction

Sentinel lymph node biopsy is the technique recommended for the axillary staging of patients with breast cancer in the initial stages without clinical axillary involvement. Three techniques are widely used globally to detect sentinel lymph nodes: patent blue, the radio-pharmaceutical technetium 99 with gamma probe, and the combination of these two.

### Objectives

To evaluate the sentinel lymph node detection rate with an innovative technique: indocyanine green (ICG) associated with fluorescence in breast cancer patients, and compare it with patent blue and a combination of patent blue and indocyanine green.

### Methods

99 patients were sequentially (not randomly) allocated into 3 arms with 33 patients submitted to sentinel lymph node techniques. One arm underwent patent blue dying, the other indocyanine green, and the third received a combination of both. The detection rates between arms were compared.

### Results

The detection rate in identifying the sentinel lymph node was 78.8% with patent blue, 93.9% with indocyanine green, and 100% with the combination. Indocyanine green identified two sentinel nodes in 48.5% of patients; the other groups more commonly had only one node identified. The mean time to sentinel lymph node identification was 20.6 ± 10.7 SD (standard deviation) minutes among patients submitted to the patent blue dye, 8.6 ± 6.6 minutes in the indocyanine green arm, and 10 ± 8.9 minutes in the combined group (P<0.001; Student's

**Funding:** RSS 1 National Council for Scientific and Technological Development (CNPq) https://www. gov.br/cnpq/pt-br The funders had no role in study design, data collection and analysis, decision to publish, or preparation of the manuscript.

**Competing interests:** The authors have declared that no competing interests exist.

test). The mean surgery time was 69.4 ± 16.9; 55.1 ± 13.9; and 69.4 ± 19.3 minutes respectively (P<0.001; Student's test).

## Conclusions

The sentinel lymph node detection rate by fluorescence using indocyanine green was 93.9%, considered adequate. The rates using patent blue, indocyanine green, and patent blue plus indocyanine green (combined) were significantly different, and the indocyanine green alone is also acceptable, since it has a good performance in sentinel lymph node identification and it can avoid tattooing, with a 100% sentinel lymph node detection rate when combined with patent blue.

## Introduction

Breast cancer treatment is a challenge to global public health as the disease is the most frequently diagnosed malignant neoplasm in the world, reaching an annual estimate of 2.26 million new cases. Moreover, it is the leading cause of cancer mortality among women. Adding to this is the worldwide alarm: two-thirds of the affected population resides in undeveloped countries [1].

The surgical treatment of breast cancer includes, in most cases, axillary surgery. Lymph node research is not indicated only in conservative surgeries in patients with ductal carcinoma *in situ* [2] or may be omitted in selected senile patients [3]. Three techniques are widely used globally to identify sentinel lymph nodes (the first node in the axilla to which malignant cells are most likely to spread through): patent blue, the radiopharmaceutical technetium 99 with gamma probe, and the combination of these two techniques. As first reported by Giuliano et al. in 1994 [4], patent blue was applied between 0.5 and 10 ml in the peritumoral region with a 25-gauge needle soon after anesthetic induction. Initially, the detection rate was 65.5% (114 out of 174 cases) [4], which was improved by Kern et al. in 1999 with rates above 98% (38 out of 40 cases) [5]. Sentinel lymph node identification using technetium 99-m (99mTc) associated with the gamma probe, described by Krag et al. in 1993, was possible with the peritumoral application of this marker 1 to 9 hours before surgery at a dose of 0.4 mCi associated with 0.5 ml of saline solution. The gamma detector for 99mTc, initially, was the C-Trak from Care Wise Medical Products (Morgan Hill, CA, USA) [6]. The detection rate of 89% was published by Borgstein et al. in 1998 [7].

Patent blue and technetium 99-m became the routine of breast surgeons, especially after the critical NSABP (National Surgical Adjuvant Breast and Bowel Project) B-32 study, which reiterated the safety of the sentinel lymph node technique, using one of these two dyes in all its 80 research centers [8]. Each of these techniques has its strengths and weaknesses. The main issue with using patent blue is the dermal tattoo, which can be eternal. Technetium 99 is expensive, and its use requires a nuclear medicine team. Keen to publicize another more modern dye option, Kitai et al. were the pioneers to report, in 2005, the technique known as ICG (indocyanine green) associated with the fluorescence image in breast cancer. The authors obtained a high identification rate of 94% [9]. ICG is a low molecular weight, non-radioactive fluorescent dye [9], a tricarbocyanine, with two heterocyclic rings joined by seven carbons, with the following molecular formula: $C_{43}H_{47}N_2NaO_6S_2$ [10]. This chemical marker can penetrate human tissue from a few millimeters to a few centimeters, allowing real-time lymphatic

migration and helping the surgeon to plan the dermal incision, thus reducing the procedure difficulty [9].

An innovative procedure raises concerns about patient safety. Patients allergic to iodine and its derivatives should refrain from ICG due to the potential risk of anaphylaxis [11]. ICG was developed in 1955 and approved for clinical use in 1959 in the United States, with an indication outside the oncological scenario. However, its use in ophthalmology for retinal angiography intensified in the 1970s. The technique of using ICG as a fluorescent marker consists of illuminating the tissue of interest at the excitation wavelength (750 to 800 nm), having filters to spectrally select the fluorescence before reaching the sensor, with more significant emissions around 800 nm and at longer wavelengths depending on concentration, pH and temperature [12].

ICG in breast surgery is still a little-used procedure [12]. Some studies have already validated ICG for the use of sentinel lymph nodes with results as good as or even better than radioisotopes. However, the main ones were published by Asian countries (China and Japan, for example, already use it on a large scale). In Europe, only a few countries developed studies in relatively small cohorts. However, this innovative technique has yet to replace the pioneers as there are not enough studies in different settings to prove the non-inferiority of ICG against patent blue and 99mTc [13].

This research presents the prospect of disseminating the results of an innovative yet little-known technique for sentinel lymph node research. The fact that the guidelines of the main oncology entities worldwide, ASCO [14] (American Society of Clinical Society), ESMO [15] (European Society of Clinical Oncology), and NCCN [16] (The National Comprehensive Cancer Network) still do not include this dye as an option in their protocols reinforces the importance of this study.

## Objectives

The primary objective of this study was to evaluate the efficacy of ICG associated with fluorescence in the breast cancer sentinel lymph node investigation. The secondary objectives were to compare the ICG technique with the patent blue and the combined method; and to evaluate the cost impact of using the ICG dye.

## Materials and methods

### Study design, setting, and ethics

This study is a non-randomized clinical trial (prospective study with a non-blinded sequential allocation of cases) performed on patients with breast cancer in initial stages (T1-T2) with clinically negative axilla who were recommended to undergo sentinel lymph node investigation during breast surgery. Patients were allocated to three groups according to the time when they were operated and the services were available locally, following the sequence: first the patent blue, then the ICG when this became available and finally the combined technique.

The Discipline of Mastology, Department of Gynecology, Escola Paulista de Medicina/Universidade Federal de São Paulo (UNIFESP) developed this research, which was carried out at Hospital de Esperança—Hospital Regional do Câncer de Presidente Prudente.

Both the Research Ethics Committee of UNIFESP (CAAE: 08169419.9.0000.5505) and the Research Ethics Committee of Hospital de Esperança—Regional Cancer Hospital of Presidente Prudente (CAAE: 08169419.9.3001.8247) approved this research protocol, following the Declaration of Helsinki. All patients submitted to the research signed the free and informed consent forms after a thorough explanation of the study and procedures to be performed. In

addition, the researchers ensured the confidentiality of all patients, not publicly disclosing the name or any other information that could identify the women involved in this study.

This study was reported according to the TREND Statement Checklist [17] and, where applicable, the CONSORT checklist [18]. The authors confirm that all ongoing and related trials for this intervention are registered at the Brazilian Clinical Trials Registry (Rebec) with the number RBR-6d36dgq. Registration could not be performed in other platforms before recruitment, as this study is not considered a clinical trial.

## Participants' selection and allocation

The study was designed with the proposed sample of 99 patients, as it is the mean sample size of the main fluorescence studies in the breast cancer scenario [9, 19, 20]. It included patients with small breast tumors in the first phases and excluded those with staging T4, N1, N2, and N3; patients who underwent neoadjuvant chemotherapy; and those with a history of allergy to iodine and its derivatives.

All patients were admitted upon arrival at the Mastology service. Patients were allocated by sequence to the three study arms between 8/2/2019 and 12/16/2021. The first 33 patients underwent sentinel lymph node biopsy using the patent blue technique. Then, patients numbers 34 to 66 were included in the ICG arm, and finally, patients 67 to 99 participated in the "combination group" that received patent blue + ICG. This non-randomized allocation was necessary because, during the beginning of the research, the surgical center used for all patients at the Regional Cancer Hospital of Presidente Prudente was the Santa Casa de Misericórdia de Presidente Prudente, an attached hospital and maintainer of the oncology hospital. The surgical center of the Regional Cancer Hospital of Presidente Prudente, which includes fluorescence technology, was inaugurated only in October/2019, a few months after the start of the research, and the full release of its use as a routine occurred only in 2020. Therefore, to maintain the sequential allocation, the second arm was the ICG arm; thus, the third and final arm was the combined arm.

## Materials used

The materials used in all cases included in this study were: patent blue dye (50 mg/2 ml), brand Pharmédice; ICG dye (5 mg/10 ml), brand OPHTALMOS; 5 ml syringes, brand SR; needle 40 x 12 (dye aspiration) and needle 30 x 8 (periareolar injection), both branded SR. For fluorescence, an image1S device coupled to a D-light P light source and a laparoscopy camera, all from the German company Karlz-Storz, were used.

## Data sources and procedures

Data were obtained by manually filling in the patient's data collection form, previously prepared with the main parameters that the study would analyze. The data were collected in a Microsoft Excel spreadsheet prepared by the statistician.

After anesthetic induction, patients in the patent blue technique group underwent periareolar injection of 2 ml of this product (a single point of application at 10 o'clock on the right breast or 2 o'clock on the left breast, applying the substance in the retro areolar region/Sappey's plexus) with subsequent five-minute breast massage for migration of the lymphotropic dye during anesthetic induction. After this period, during the axillary surgery, the lymphatic ducts were visualized up to the sentinel lymph node (the first lymph node of the lymphatic drainage of the breast), which was resected.

Patients in the ICG technique group underwent periareolar injection of 5 mg/10 ml of this product (a single point of application at 10 o'clock on the right breast or 2 o'clock on the left

breast, applying the substance in the retro areolar region/Sappey's plexus), followed by a five-minute breast massage for migration of the fluorescent dye, also immediately following anesthetic induction. After this period, in the axillary surgery, the lymphatic ducts up to the sentinel lymph node were visualized using a Karlz-Storz fluorescence device called image 1S, coupled to a D-light P light source and a camera of real-time visualization that has infrared light with 760 nm, coupled to a laparoscopy fluorescence camera. The initial idea would be to use the device that is specific for open surgery called VITOM II ICG. However, The Hospital did not acquire this equipment (there was a purchase negotiation by the Hospital's Engineering, but the high cost made it impossible). Thus, we used the fluorescence camera for laparoscopy, a device that the Regional Cancer Hospital of Presidente Prudente already had. There was no damage to image quality since this device performs fluorescence in closed surgeries, and in the literature, there is already a description of its use also in open breast surgeries [17].

The patients in the group with the combined technique underwent the two procedures as previously described, first the ICG and then patent blue. Axillary procedures were performed according to the ACOSOG (American College of Surgeons Oncology Group) Z0011 protocol [20].

One of the three breast surgeons on the hospital's team performed the procedures. The access route for identifying the sentinel lymph node was the best from an aesthetic point of view. Thus, some patients had only one breast incision (superior lateral quadrant, for example), and others had two breast incisions. The researchers did not try to modify the surgeon's routine and preference in this regard.

## Variables

During surgery we analyzed:

- The time to identification of the sentinel lymph node (the first dermal incision was considered the onset time). We used a digital clock on the wall of the operating room to measure the time;

- The number of lymph nodes identified;

- Total surgery time (procedure time was considered from the first dermal incision to the end of the last dermal suture). The same digital clock in the surgical procedure room was used to measure time accurately;

- Intraoperative complications;

- BMI (body mass index).

We analyzed the measurements of the scar(s) on the seventh postoperative day and the presence or absence of a dermal tattoo on the 60th postoperative day.

Axillary status was defined according to the number of positive lymph nodes. The detection rate was defined as the number of patients with sentinel lymph nodes identified by each of the three methods divided by the total number of patients.

Tumor grade was classified using the modified Bloom-Richardson system [21], and staging was performed using the AJCC Cancer Staging Manual 8th Edition [22]. In the immunohistochemistry evaluation, the cutoff value for the Ki-67 was a 14% score, and the variable was analyzed categorically as above or below this reference score. The immunohistochemical profile was classified into Luminal B, Luminal A, Luminal HER, Triple Negative, and HER 2+.

The costs involved in the treatments at Hospital de Esperança for each patient underwent a study of the economic impact of the procedure. The Hospital's Financial Department analyzed

all the amounts paid during the entire hospitalization: fees, food, medication, serum therapy, pathology, breast implants, and the dyes used (with values expressed in the local currency, Brazilian reais, BRL).

## Statistical analysis

Data were presented in terms of their frequencies and percentages for categorical variables, means and standard deviations (SD) for continuous variables, and medians (minimum-maximum) for discrete variables. In comparing the variables between the groups of dyes, analysis of variance (ANOVA) was used for normally distributed variables, followed by the Tukey test as post-hoc; the Kruskal-Walis test for non-normally distributed variables, and the Chi-Square test (or Fisher's exact test when appropriate) for categorical variables. The ANOVA and Tukey tests were used to analyze the costs according to the type of dye on multiple comparisons. Patients who underwent the breast reconstruction technique with the expander were excluded from this economic analysis due to the high cost of the material.

P-values less than 0.05 were considered statistically significant. The R software, version 4.1.3, was used in the analysis.

## Results

No patient was lost to follow-up during the research period. The patients' flowchart is represented in Fig 1. The total N proposed for the study of 99 patients was met, with three arms with 33 patients each.

The mean age of patients was 58.4 years (SD of 13.3). During the study period, there were no male patients or patients with ductal carcinoma *in situ*. The vast majority of patients, 70.7% (n = 70), were already postmenopausal, and the remainder (29.3%, n = 29) were still in premenopausal (Table 1). Most patients were overweight (44; 44.4%) or obese (28; 28.3%) during the evaluation of the first medical appointment.

The surgeons performed breast-conserving surgery in 84.8% (n = 84), mastectomy without breast reconstruction in 10.1% (n = 10), and mastectomy with breast reconstruction (expander) in 5.1% (n = 5). The histological subtypes had the distribution shown in Table 1,

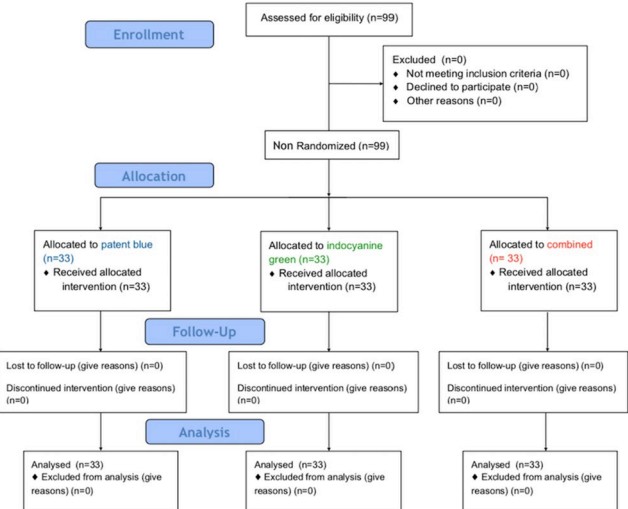

**Fig 1. Flowchart of patient inclusion in the study.**

**Table 1. Characteristics of patients and tumors in the three study arms.**

| Characteristic | n = 99 (%) |
|---|---|
| **Age** | |
| Mean ± standard deviation | 58.4 ± 13.3 |
| **Menopausal status** | |
| Premenopausal | 29 (29.3%) |
| Postmenopausal | 70 (70.7%) |
| **BMI** | |
| 18.5–24.9 | 27 (27.3%) |
| 25–29.9 | 44 (44.4%) |
| ≥ 30 | 28 (28.3%) |
| **Breast region** | |
| Right inferior lateral quadrant | 2 (2.0%) |
| Left inferior lateral quadrant | 5 (5.0%) |
| Left inferior medial quadrant | 1 (1.0%) |
| Right superior lateral quadrant | 21 (21.2%) |
| Left superior lateral quadrant | 14 (14.1%) |
| Right superior medial quadrant | 7 (7.1%) |
| Left superior medial quadrant | 6 (6.1%) |
| Right retro areolar region | 4 (4.0%) |
| Left retro areolar region | 8 (8.1%) |
| Union of the lateral quadrants of the right breast | 6 (6.1%) |
| Union of the upper quadrants of the right breast | 7 (7.1%) |
| Union of the upper quadrants of the left breast | 7 (7.1%) |
| Union of the lower quadrants of the right breast | 1 (1.0%) |
| Union of the lower quadrants of the left breast | 2 (2.0%) |
| Union of the lateral quadrants of the left breast | 8 (8.08%) |
| **Type of surgery** | |
| Breast-conserving surgery (quadrantectomy) with sentinel lymph node biopsy | 84 (84.8%) |
| Mastectomy with sentinel lymph node biopsy with breast reconstruction (expander) | 5 (5.1%) |
| Mastectomy with sentinel lymph node biopsy without reconstruction | 10 (10.1%) |
| **Anatomical staging** | |
| IA | 53 (53.5%) |
| IIA | 27 (27.3%) |
| IIB | 12 (12.1%) |
| IIIA | 4 (4.0%) |
| IIIC | 3 (3.0%) |
| **Prognostic staging** | |
| IA | 73 (73.7%) |
| IB | 12 (12.1%) |
| IIA | 10 (10.1%) |
| IIB | 12 (12.1%) |
| IIIB | 2 (2.0%) |
| **Tumor size** | |
| 6-10mm (T1b) | 13 (13.1%) |
| 11-20mm (T1c) | 53 (53.5%) |
| 21-50mm (T2) | 33 (33.3%) |
| **Histology** | |
| Invasive carcinoma of no special type | 92 (92.9%) |

(*Continued*)

**Table 1.** (Continued)

| Characteristic | n = 99 (%) |
|---|---|
| Invasive lobular carcinoma | 4 (4.0%) |
| Invasive medullary carcinoma | 1 (1.0%) |
| Invasive papillary carcinoma | 2 (2.0%) |
| **Grade** | |
| 1 | 5 (5.1%) |
| 2 | 64 (64.6%) |
| 3 | 30 (30.3%) |
| **Immunohistochemistry** | |
| HER2+ | 7 (7.1%) |
| Luminal A | 37 (37.4%) |
| Luminal B | 35 (35.4%) |
| Luminal HER | 12 (12.1%) |
| Triple Negative | 8 (8.1%) |

with the most common as non-special invasive carcinoma 92.9% (n = 92). The tumor size was classified between T1b, T1c, and T2, and T1c was the most common (53.5%; n = 53). No cases of T1a and T3 were reported. Most patients were Luminal A or B in the immunohistochemical profile.

The number of lymph nodes identified in most cases was one or two (Table 2). Nine cases in total had no lymph nodes identified by the techniques used (Table 2).

Combining all the study arms, the mean time for sentinel lymph node identification was 12.7 minutes, and the total surgery time was 64.6 minutes. The majority of patients (55.6%) received only one breast incision. The mean size of the breast incision was 8.3 cm when the incision was single. However, when two incisions were performed, the mean measurement of the breast incision was 6.1 cm, and the axillary incision was 5.5 cm.

There were no intraoperative complications, and the dermal tattoo occurred in 34 cases, corresponding to 56.7% of the patients.

There was no statistically significant difference between the study arms except for tumor size (Table 3), although T1c was the main stage classified in all groups.

The accuracy rate in identifying the sentinel lymph node was 78.8% with patent blue, 93.9% with ICG, and 100% with patent blue + ICG (Fig 2). The combined group identified mainly two sentinel nodes (48.5%); however, the other groups more commonly identified one sentinel node only (Table 4).

There were no significant differences (p = 0.850) between patients with or without lymph nodes identified regarding age (58.6 ± 13.6 years versus 57.2 ± 10.8 years respectively) or BMI (28.1 ± 5.2 kg/m2 and 28.5 ± 7.2 kg/m2).

The mean time to sentinel lymph node identification was 20.6 minutes among patients submitted to the patent blue dye, 8.6 minutes in the ICG arm, and 10 minutes in the combination of the two methods (P<0.001; Student's test for independent samples). The mean surgery time was 69.4 minutes with patent blue, 55.1 minutes with ICG, and 69.4 minutes with the combination (P< 0.001; Student's test) (Fig 3).

The difference in the measurement of the breast incision was statistically significant only in the cases where a single incision was used (P = 0.020), with the smallest incision, 6.4 cm, in the ICG arm. The analysis of the measurement of the breast or axillary incision when performed in association, that is, two incisions, did not reveal significant differences (Table 4).

**Table 2. Surgical aspects of the three methods for sentinel lymph node detection.**

| Characteristic | n = 99 (%) |
|---|---|
| **Dye** | |
| Patent Blue | 33 (33.3%) |
| Indocyanine Green | 33 (33.3%) |
| Patent Blue + Indocyanine Green | 33 (33.3%) |
| **Number of lymph nodes identified** | |
| 0 | 9 (9.2%) |
| 1 | 46 (46.9%) |
| 2 | 35 (35.7%) |
| 3 | 7 (7.1%) |
| 4 | 1 (1.0%) |
| Median (min—max) | 1 (0–4) |
| **Identification time (min)** | |
| Mean ± SD | 12.7 ± 10.1 |
| **Surgery time (min)** | |
| Mean ± SD | 64.6 ± 18.0 |
| **Intraoperative complications** | |
| Dye allergy | 2 (2.0%) |
| **Number of incisions** | |
| 1 | 55 (55.6%) |
| 2 | 44 (44.4%) |
| **Breast incision measurement (cm)** | |
| Mean ± SD | 6.1 ± 1.4 |
| **Axillary incision measurement (cm)** | |
| Mean ± SD | 5.5 ± 8.8 |
| **Single breast incision measurement (cm)** | |
| Mean ± SD | 8.3 ± 3.9 |
| **Tattoo on breast skin at follow up** | 34 (56.7%) |

When comparing the groups, the main postoperative adverse event was the dermal tattoo, represented in 24 cases (72.7%) in the patent blue group (P = 0.002). Among all patients with a dermal tattoo, 71% corresponded to the patent blue arm (Fig 4).

The cost of surgeries performed with the patent blue dye and the combined arm were not significantly different. However, the cost of surgeries performed with the green dye (ICG) was significantly lower than both. The mean total costs are shown in Table 5. Although the product ICG has a higher cost (BRL 150.00), the cost of hospitalization of patients submitted to this dye was lower than that of the patent blue dye p = 0.013 (BRL 40.00), and the combined groups p = 0.003 (BRL 150.00 + BRL 40.00 = BRL 190.00). The patent blue and combined arms costs were not statistically different (p = 0.905), as shown in Fig 4.

Table 6 summarises the number of lymph nodes identified in the group receiving the combined intervention.

## Discussion

In this study, the detection rate was surprisingly higher in patients who used fluorescence, reaching 100% when combined with patent blue. This real data on a new technique for sentinel lymph nodes may allow cancer centers that do not have a nuclear medicine service in the future to consider getting access to another option in addition to the renowned patent blue,

**Table 3. Comparison of the characteristics of patients and tumors in the three study arms.**

| Characteristic | Patent blue (n = 33) | Patent blue + ICG (n = 33) | ICG (n = 33) | P-value[1] |
|---|---|---|---|---|
| **Age** | | | | |
| Mean ± standard deviation | 58.7 ± 13.9 | 59.6 ± 13.3 | 57.0 ± 13.0 | 0.717 |
| Premenopausal | 11 (33.3%) | 6 (18.2%) | 12 (36.4%) | 0.220 |
| Postmenopausal | 22 (66.7%) | 27 (81.8%) | 21 (63.6%) | |
| **BMI** | | | | |
| 18.5–24.9 | 10 (30.3%) | 7 (21.2%) | 10 (30.3%) | 0.875 |
| 25–29.9 | 15 (45.5%) | 16 (48.5%) | 13 (39.4%) | |
| $\geq 30$ | 8 (24.2%) | 10 (30.3%) | 10 (30.3%) | |
| **Region** | | | | |
| Right inferior lateral quadrant | 0 (0%) | 1 (3%) | 1 (3%) | **0.026** |
| Left inferior lateral quadrant | 2 (6.1%) | 2 (6.1%) | 1 (3%) | |
| Left inferior medial quadrant | 0 (0%) | 0 (0%) | 1 (3%) | |
| Right superior lateral quadrant | 11 (33.3%) | 5 (15.2%) | 5 (15.2%) | |
| Left superior lateral quadrant | 4 (12.1%) | 3 (9.1%) | 7 (21.2%) | |
| Right superior medial quadrant | 2 (6.1%) | 4 (12.1%) | 1 (3%) | |
| Left superomedial quadrant | 1 (3%) | 1 (3%) | 4 (12.1%) | |
| Right retro areolar region | 2 (6.1%) | 2 (6.1%) | 0 (0%) | |
| Left retro areolar region | 4 (12.1%) | 2 (6.1%) | 2 (6.1%) | |
| Union of the lateral quadrants of the right breast | 1 (3%) | 3 (9.1%) | 2 (6.1%) | |
| Union of the upper quadrants of the right breast | 2 (6.1%) | 1 (3%) | 4 (12.1%) | |
| Union of the upper quadrants of the left breast | 2 (6.1%) | 5 (15.2%) | 0 (0%) | |
| Union of the lower quadrants of the right breast | 0 (0%) | 1 (3%) | 0 (0%) | |
| Union of the lower quadrants of the left breast | 2 (6.1%) | 0 (0%) | 0 (0%) | |
| Union of the lower quadrants of the left breast | 0 (0%) | 3 (9.1%) | 5 (15.2%) | |
| **Type of surgery** | | | | |
| Breast-conserving surgery (quadrantectomy) with sentinel lymph node biopsy | 27 (81.8%) | 27 (81.8%) | 30 (90%) | 0.555 |
| Mastectomy with sentinel lymph node biopsy with breast reconstruction (expander) | 2 (6.1%) | 3 (9.1%) | 0 (0%) | |
| Mastectomy with sentinel lymph node biopsy without breast reconstruction | 4 (12.1%) | 3 (9.1%) | 3 (9.1%) | |
| **Anatomical staging** | | | | |
| IA | 17 (51.5%) | 14 (42.4%) | 22 (66.7%) | 0.063 |
| IIA | 7 (21.2%) | 12 (36.4%) | 8 (24.2%) | |
| IIB | 8 (24.2%) | 2 (6.1%) | 2 (6.1%) | |
| IIIA | 0 (0%) | 3 (9.1%) | 1 (3.0%) | |
| IIIC | 1 (3.0%) | 2 (6.1%) | 0 (0%) | |
| **Prognostic staging** | | | | |
| IA | 19 (57.6%) | 27 (81.8%) | 27 (81.8%) | 0.090 |
| IB | 8 (24.2%) | 1 (3.0%) | 3 (9.1%) | |
| IIA | 5 (15.2%) | 3 (9.1%) | 2 (6.1%) | |
| IIB | 0 (0%) | 1 (3.0%) | 1 (3.0%) | |
| IIIB | 1 (3.0%) | 1 (3.0%) | 0 (0%) | |
| **Tumor size** | | | | |
| 6-10mm (T1b) | 1 (3%) | 4 (12.1%) | 8 (24.2%) | **0.046** |
| 11-20mm (T1c) | 18 (54.5%) | 16 (48.5%) | 19 (57.6%) | |
| 21-50mm (T2) | 14 (42.4%) | 13 (39.4%) | 6 (18.2%) | |
| **Histology** | | | | |

(*Continued*)

**Table 3.** (Continued)

| Characteristic | Patent blue (n = 33) | Patent blue + ICG (n = 33) | ICG (n = 33) | P-value[1] |
|---|---|---|---|---|
| Invasive carcinoma of no special type | 31 (93.9%) | 29 (87.9%) | 32 (97.0%) | 0.346 |
| Invasive lobular carcinoma | 1 (3.0%) | 3 (9.1%) | 0 (0%) | |
| Invasive medullary carcinoma | 0 (0%) | 0 (0%) | 1 (3.0%) | |
| Invasive papillary carcinoma | 1 (3.0%) | 1 (3.0%) | 0 (0%) | |
| **Grade** | | | | |
| 1 | 1 (3.0%) | 1 (3.0%) | 3 (9.1%) | 0.786 |
| 2 | 22 (66.7%) | 22 (66.7%) | 20 (60.6%) | |
| 3 | 10 (30.3%) | 10 (30.3%) | 10 (30.3%) | |
| **Immunohistochemistry** | | | | |
| HER2+ | 1 (3.0%) | 4 (12.1%) | 2 (6.1%) | 0.377 |
| Luminal A | 13 (39.4%) | 11 (33.3%) | 13 (39.4%) | |
| Luminal B | 15 (45.5%) | 12 (36.4%) | 8 (24.2%) | |
| Luminal HER | 2 (6.1%) | 5 (15.1%) | 5 (15.1%) | |
| Triple Negative | 2 (6.1%) | 1 (3.0%) | 5 (15.1%) | |

1 p-values for the ANOVA test for all continuous variables (normality was checked with the Shapiro-Wilk test) and the Chi-Square test for all categorical variables. The letters followed by the p-values represent a statistically significant result for the Anova post-hoc test, where a: patent blue x patent blue + ICG, b: patent blue x ICG, c: patent blue + ICG x ICG. P-values in bold were statistically significant at the level $\alpha$ = 5%.

which most hospitals still use, especially in public health. Detection rates with ICG reported in other publications are also high. Kitai et al. [9], a pioneer in its use, obtained a rate of 94% in Japan in 2004 (18 patients). Hirche et al. [23] reached 97.7% in Germany in 2010 (n = 43), and Sorrentino et al. [19] 92.7% in Italy in 2018 (n = 70).

Despite the high detection rates of sentinel lymph node biopsy using ICG, the systematic review of 15 studies in this scenario, published in 2014 by Ahmed et al. [24], could not identify

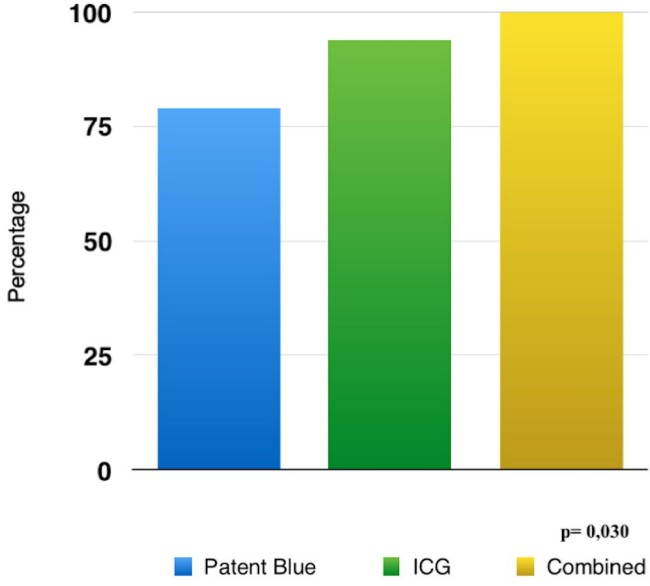

**Fig 2. Sentinel lymph node identification rate.**

**Table 4. Comparison of surgical aspects of the three methods for sentinel lymph node detection.**

| Characteristic | Patent blue (n = 33) | Patent blue + ICG (n = 33) | ICG (n = 33) | P-value[1] |
|---|---|---|---|---|
| **Number of lymph nodes identified** | | | | |
| 0 | 7 (21.2%) | 0 (0%) | 2 (6.1%) | |
| 1 | 16 (48.5%) | 18 (54.5%) | 12 (36.4%) | |
| 2 | 7 (21.2%) | 12 (36.4%) | 16 (48.5%) | **0.030** |
| 3 | 2 (6.1%) | 2 (6.1%) | 3 (9.1%) | |
| 4 | 1 (3.0%) | 0 (0%) | 0 (0%) | |
| Median (min—max) | 1.0 (0.0–4.0) | 2.0 (0.0–3.0) | 1.0 (1.0–3.0) | 0.056 |
| **Identification time** | | | | |
| (min) | | | | |
| Mean ± SD | 20.6 ± 10.7 | 10.0 ± 8.9 | 8.6 ± 6.6 | <**0.001**a,b |
| **Surgery time (min)** | | | | |
| Mean ± SD | 69.4 ± 16.9 | 69.4 ± 19.3 | 55.1 ± 13.9 | <**0.001**b,c |
| **Intraoperative complications** | | | | |
| Dye allergy | 1 (3.0%) | 1 (3.0%) | 0 (0%) | - |
| **Number of incisions** | | | | |
| 1 | 19 (57.6%) | 19 (57.6%) | 17 (51.5%) | |
| 2 | 14 (42.4%) | 14 (42.4%) | 16 (48.5%) | 0.849 |
| **Breast incision measurement (cm)** | | | | |
| Mean ± SD | 6.4 ± 1.5 | 5.7 ± 0.9 | 6.3 ± 1.5 | 0.333 |
| **Axillary incision measurement (cm)** | | | | |
| Mean ± SD | 4.5 ± 0.9 | 4.0 ± 1.1 | 7.5 ± 14.4 | 0.493 |
| **Single breast incision measurement(cm)** | | | | |
| Mean ± SD | 10.0 ± 5.1 | 8.3 ± 3.2 | 6.4 ± 1.6 | **0.020**b |
| **Tattoo on breast skin at follow up** | 24 (72.7%) | 10 (30.3%) | 0 | **0.002** |

1 p-values for the ANOVA test for all continuous variables (normality was checked with the Shapiro-Wilk test) and the Chi-Square test for all categorical variables. The letters followed by the p-values represent a statistically significant result for the Anova post-hoc test, where a: Patent Blue x Patent Blue + ICG, b: Patent Blue x ICG, c: Patent Blue + ICG x ICG. P-values in bold were statistically significant at the level α = 5%.

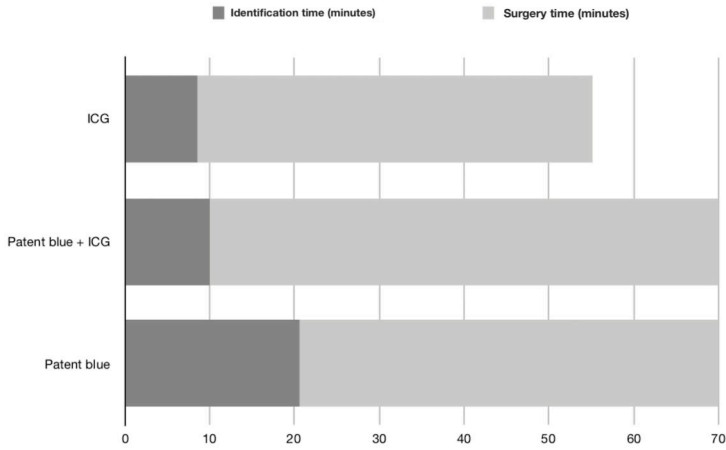

**Fig 3. Mean time for identification and total surgery time according to the technique used for sentinel lymph node identification.**

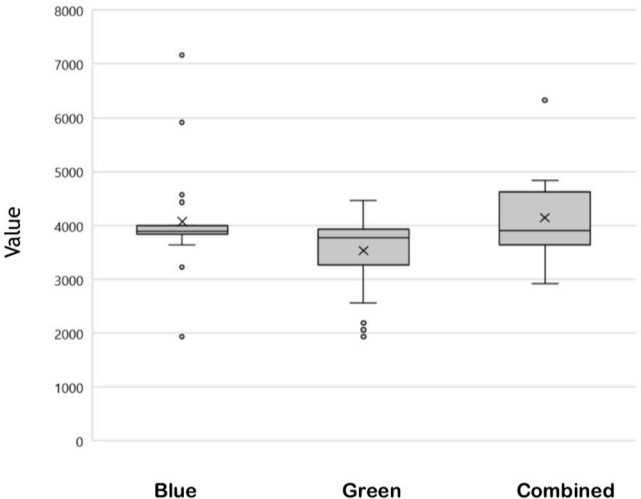

**Fig 4. Comparison between the dye values.**

a significant benefit when adopting this new technique in comparison with the two standard techniques (patent blue and technetium). The authors suggested the need for further refinement of the ICG technique through the performance of more robust trials. In contrast, Zhang et al. [25] published a meta-analysis in 2016 that jointly evaluated a total of 2594 patients, which obtained a combined sensitivity of 0.92 (95% CI, 0.85–0 .96), specificity of 1 (95% CI, 0.97–1) and DOR (Diagnostic Odds Ratio) of 311.47 (95% CI, 84.11–1153.39). The authors concluded that the ICG technique is indeed viable for the identification of the sentinel lymph node of patients with breast cancer with clinically negative axilla, also recommending further studies such as multicenter clinical trials and with longer follow-ups.

The combined technique, ICG + patent blue, has already been tested by Shen et al. [26] in China and published in 2018 with a detection rate of 99.2%. A factor that could impact the detection rate would be the difference between the patients' BMI or age; however, the distribution was homogeneous between the groups in our study (BMI P = 0.875, age P = 0.220), with overweight (BMI between 25–29,9) and postmenopausal patients prevailing.

The dose of patent blue application is already consolidated in clinical practice, with the entire dye ampoule usually infiltrated (5 mg/2 ml). However, even the main study, published by Giuliano et al., which described the use of the technique in breast oncology, showed a divergence between the doses received by the patients: the first 20 patients received doses of 0.5 to 10 ml. However, from patient 21 to patient 172, the standard dose of the established dye was

**Table 5. Descriptive summary of costs by type of dye in Brazilian reais (BRL).**

| Dye | Minimum | Mean | Maximum | Sum | Standard Deviation | P-value* |
|---|---|---|---|---|---|---|
| Blue | 1,930.83 | 4,068.34 | 7,170.70 | 134,255.20 | 870.91 | **0.004**[b,c] |
| Green | 1,937.83 | 3,532.11 | 4,459.17 | 116,559.60 | 604.83 | |
| Combined | 2,916.94 | 4,147.20 | 6,371.44 | 136,857.60 | 753.81 | |

* P-value referring to the ANOVA test. Letters in superscript referring to multiple comparisons by the Tukey test, where b: patent blue x ICG, c: patent blue + ICG x ICG. The P-value in bold was statistically significant at the level α = 5%.

**Table 6. Number of lymph nodes identified in the combined arm.**

| Dye visualized | Number of lymph nodes identified |
|---|---|
| Blue | 0 |
| Green | 10 (30.4%) |
| Both blue and green | 23 (69.6%) |

between 3 to 5 ml, which evidences the difficulty of determining the ideal dose of a new technique recently incorporated by the surgeon [12].

The acceptable SNLB identification rate in the literature is 90% or higher. However, in our research, the patent blue group had a lower rate (78.8%). Patients who did not have SNLB identification were submitted to Berg's axillary level I sampling resection. We cannot explain the low accuracy rate of this dye, especially considering that the three surgeons on the team are trained in using patent blue [27].

Lumpectomy was the main type of surgical treatment performed in patients in this study in all arms (81.8% in patent blue and combined; 90.9% in ICG). Access to adjuvant radiotherapy allows this good rate of conservative surgery and reiterates the global trend of performing mastectomy only when precisely indicated and necessary [28]. Some patients who underwent mastectomy did not want immediate reconstruction (10.1% adding the three groups), even after the offer, explanation, and encouragement. Our service has a team of reconstructive surgery and easy access to breast implants. However, the patient's wishes were always respected. All cases that underwent immediate breast reconstruction (5.1%) underwent the expander technique. The different types of dyes for sentinel lymph node detection did not change the surgeon's approach (type, location, or the number of incisions).

The dose of ICG application and the camera used for fluorescence detection is still heterogeneous in the literature, but the injection has always been described in the periareolar region. The pioneering Japanese study in the procedure used a dose of 25 mg/5 ml of the Diagnogreen 0.5% brand (Daiichi Pharmaceutical, Tokyo Japan), infiltrated soon after anesthesia, and performed a two-minute breast massage, detected with the device from Hamamatsu Photonics (Japan) [9]. The German study applied the ICG Solution at 11 mg, with massage for 5 to 15 minutes, with the IC-View, Pulsion Medical Systems (Munich, Germany) [23]. The Italian study, in turn, drastically reduced the dose to 1.5 mg of the ICG-Pulsion dye (Pulsion Medical Systems, Munich, Germany) with the D-Light P 20cm camera coupled with the image1 S Camera Platform (Karl-Storz, Germany)—which was created for laparoscopy but can also be used for open surgery [19]. No details are described about whether the massage was performed. This material used in the Italian study is similar to the one used in this research and shares the characteristic of using a single system for different surgeries. The Chinese study evaluated three arms, similar to this research, and used a dose of 6.5 mg with the Photodynamic eye camera (Hamamatsu Photonics, Japan) with a 10-minute massage [26]. We emphasize the difficulty in determining the exact time of massage, which is described in a heterogeneous way by the literature, even in the scenario of the classic patent blue: Giuliano et al. performed an interval of 1 to 20 minutes between the first 20 cases. From the 21st patient onwards, the standard time was defined as 5 minutes, which most oncology services currently follow [4, 12].

A crucial detail for the quality and safety of the ICG technique is the dilution of this substance. Yamagami et al. [29] described that increasing the ICG concentration may actually decrease fluorescence, because fluorophos exhibit fluorescence suppression at very high concentrations, with no benefit in applying a high concentration for sentinel lymph node biopsy. Careful application of the most diluted ICG is a factor that improves the performance of this

technology. This same Japanese group also described the so-called "extinction reaction" by diluting the ICG 100 x in saline solution [1.0 ml (0.05 mg)] when applied in combination with blue or 2.0 ml (0.1 mg) when applied alone. Another valuable tip described by this author is using a needle smaller than the 27 gauge to increase the pressure in the infiltration. In our study, we used the total content of the smallest dose available in Brazil: 5.0 mg that comes in a powdered bottle and 10 ml of distilled water. As it was the first contact with this technique, we used the dose and dilution proposed in the research project following the main pioneering studies of this technique, which also paid more attention to dosage than to dilution. That is, the ICG technique works well regardless of the dose applied, but a small dose and a higher dilution (100 x) may be sufficient and provide a better visual experience of fluorescence.

This study was a pioneer in comparing the time for sentinel lymph node identification among the dyes used. The advent of fluorescence also allowed less time to visualize the first lymph node of the axillary drainage and its excision (12-minute reduction compared to patent blue). Therefore, there was a reduction in the mean duration of the surgical procedure in relation to other dyes (14.3-minute reduction). Surgery time was similar between the patent blue and combined arms. The main benefit of reducing sentinel lymph node identification time and surgery time is the patient's shorter exposure time to general anesthesia and mechanical ventilation, which may reduce the risk of intraoperative and postoperative complications and the lower cost of anesthetic drugs.

Despite the research, the team of surgeons focused on keeping their surgical indications the same and maintaining the same type, quantity, and topography of the incisions. The sentinel lymph node technique is classically performed through an additional axillary incision [26]. However, breast surgeons currently aim for better cosmetic results whenever possible, and some patients receive only a dermal scar for breast cancer and axillary surgery [30]. Following this trend, more than half of the cases received only one breast incision. The three groups were homogeneous regarding the number of incisions (p = 0.849). The single incision had a smaller measurement in the ICG group, 3.4 cm less than with patent blue alone and 1.9 cm less than the combined technique. This statistically significant difference (p = 0.020) may result in a lesser breast mutilation aspect in the postoperative period and better cosmesis.

We realized that the ICG is ultrasensitive and migrates through the lymphatic pathway extremely quickly (Fig 5). However, in some cases, we observed that the dye leaked into adjuvant tissues to the lymph nodes. Thus, we believe a lower dose can be sufficient and efficient. We strictly maintained the massage dose and time proposed in the research project to guarantee research homogeneity. We hope that further studies can determine the ideal dose, even encouraging the pharmaceutical industry to market vials with a specific presentation for the axillary lymph node scenario. Our suggestion is to test the 1 mg dose for breast application, which seems to be sufficient.

Indeed, when evaluating a procedure that is still little known and used among breast surgeons, in addition to its accuracy, we must be concerned about its safety. Hua et al. [31] recently published data that corroborate our research findings. 194 Chinese patients underwent ICG associated with fluorescence, and none had intraoperative complications or dermal tattooing. Even more impressive data from this study revealed only 2 cases of recurrence at 67-month follow-up with disease-free and 5-year overall survival of 92.3% and 98.3%, respectively. To study this possible mechanism, we hope to publish a graph of disease-free survival and overall survival from the 60-month follow-up of patients participating in this research soon. Asaga et al. published in 2021 a study with 565 patients with an even longer follow-up of 83 months, demonstrating results with a favorable prognosis (disease-free survival of 92.4% and overall survival of 97.3% and reiterating that the ICG technique is an alternative to the use of technetium in patients with breast cancer with clinically negative armpits [32]. Wang et al.

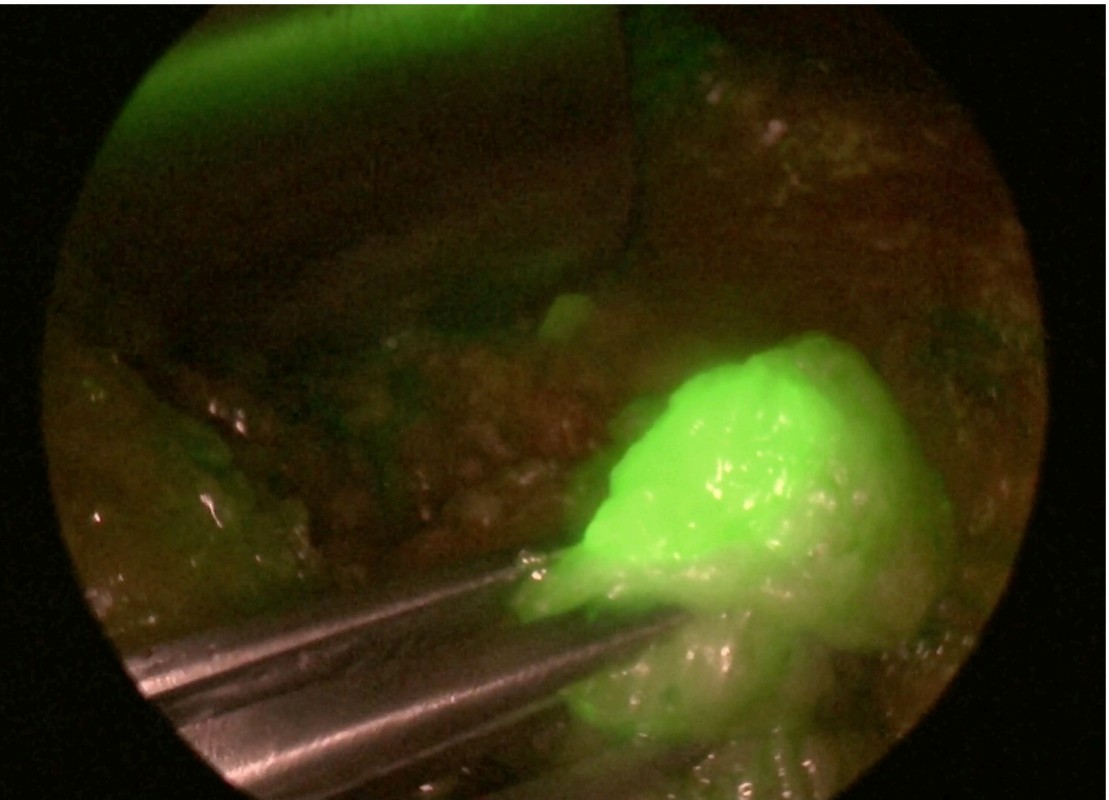

**Fig 5. Sentinel lymph node with fluorescent image after application of indocyanine green dye.**

[33] published, also in 2021, a study with an even larger number of patients evaluated (969) and a mean follow-up of 5.6 years (2–9.3 years) reporting only 0.64% of axillary recurrence.

None of the 66 patients who used ICG (33 in the isolated arm and 33 in the combined arm) in our study presented allergy, reiterating that atopy is extremely rare. However, for safety, we investigated allergies to iodides at anamnesis, preparing the anesthesiologist for a possible reversal of the atopic condition if necessary. The dermal tattoo caused by patent blue does not interfere with the oncological treatment but generates visual discomfort. More than two-thirds of the cases of dermal tattoos were caused by patent blue (71%, p = 0.002) (Fig 6).

The main challenge for our team was to start a new technique without the help of another surgeon with experience in the ICG because, in our country, this dye is not yet used for axillary SNLB. Nevertheless, we kept the recruitment running during the COVID-19 pandemic since many patients with luminal tumors were initially submitted to neoadjuvant endocrine therapy to optimize the operating room's use.

Our device is still at the forefront of this technology. However, new technologies have been incorporated for commercialization, and today there is already a more modern device, also from Karl-Storz, called IMAGE1 S RUBINA, which already enables 3D and 4K images [34]. We know that not all hospitals have this fluorescence equipment, which is still expensive. However, with the advent of robotics, some of these devices already include fluorescence technology [35]. In addition to consolidating national data on using the ICG, we aim to disseminate it widely to other breast and oncological surgery services so that more patients and surgeons can also benefit from learning this technique. For hospitals that have not yet had access to the high technology of fluorescence, an option in the near future is the

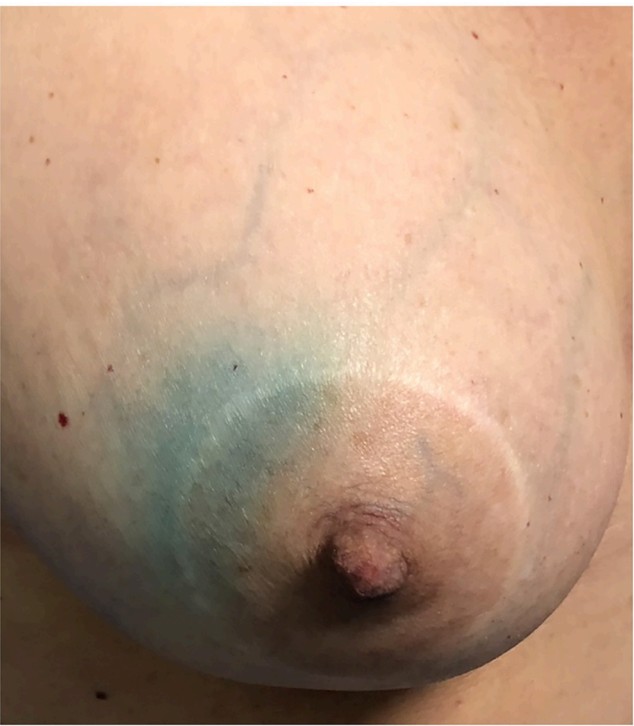

**Fig 6. Periareolar dermal tattoo of patent blue dye present in a patient at the 60th-day of follow-up.**

commercialization of fluorescence devices of national manufacture, which are still prototypes and have been tested in head and neck surgery at Hospital de Amor (Barretos—São Paulo) with promising results [36].

Three valuable recommendations (which we learned after some surgeries) that we make to start using the ICG technique:

- Perform effective breast massage after dye application.

- Turn off the operating room light for better visualization of the fluorescence image on the monitor.

- Perform lymph node investigation as soon as possible to avoid dye extravasation to the axillary wall.

The ICG has also been effectively used in other gynecological tumors, such as the endometrium and cervix. In the endometrial oncology scenario, the FIRES study (*Sentinel lymph node biopsy to lymphadenectomy for endometrial cancer staging*) showed a low false negative rate, only 3%, when identifying lymph node metastasis during sentinel lymph node biopsy [37]. The FILM study (near-infrared fluorescence for detection of sentinel lymph nodes in women with cervical and uterine cancers) revealed a higher rate of identification of the number of sentinel lymph nodes resected using ICV compared to blue (97% x 47%) also in patients with cervical cancer [38]. Despite being important data for the respective specialties, it is of little use in breast oncology due to the great impact of ACOSOG Z011 [20] and AMAROS [39]. These protocols strengthened the safety of radiotherapy in patients with less than three positive axillary lymph nodes, conservative surgery, and mastectomy, who no longer undergo complete axillary lymph node dissection.

Our study found a statistically significant difference between the number of sentinel lymph nodes found between the three groups. The ICG technique found two lymph nodes in most cases (48.5%). In the patent blue and combined groups, only one lymph node was found in most cases (48.5% and 54.5% respectively). We don't have an exact explanation for this finding. A possible theory would be the fact that ICG has a molecular weight (774.96 g/mol) greater than that of patent blue (566.66 g/mol) and therefore when used alone, stains only one main lymph node (single sentinel lymph node). An analysis with a larger number of patients could confirm this finding. In terms of the effectiveness of surgical treatment, this finding has no clinical significance, since it is already well established that only the stained lymph node(s) should be resected during sentinel lymph node biopsy in primary surgery for breast cancer [20, 40]. In sentinel lymph node biopsy after neoadjuvant chemotherapy, the recommendation is to use double labeling and to resect at least 3 lymph nodes, to obtain a false negative rate below 10% [42].

Therapeutic effects of ICG have also been rudimentarily reported, such as in the treatment of acne and metastatic nodules [10, 41]. Shen et al. published that there is no statistically significant difference in disease-free survival at 50 months of follow-up in patients submitted to patent blue (n = 149) and patent blue + ICG (n = 374), p = 0.322 [26].

The application of ICG associated with fluorescence has been tested for SLNB after neoadjuvant chemotherapy. A prospective study included patients with an initially clinically positive single axillary lymph node (cN1) and registered a lymph node detection rate of 89.2% with the modern dye versus 85.5% using traditional patent blue [42]. These data strengthen the versatility of the ICG in lymph node research.

The monetary value of hospitalization for each patient in the study was performed to confirm that the cost of a more modern dye would not financially impact the institution. One reason that explains the unusual finding in comparison with patent blue is that, by chance, breast reconstruction was not performed in patients who used ICG. The cost of materials such as breast implants and the fees for breast reconstruction contribute to the overall cost of hospitalization. It is important to emphasize that the cost of the fluorescence equipment is high; however, this equipment already belonged to the structure of the surgical center of Hospital de Esperança before the research started, and we had no additional costs for using this technology.

## Conclusions

The sentinel lymph node detection rate by fluorescence using indocyanine green was 93.9%, considered adequate. The rates using patent blue, ICG, and patent blue plus ICG (combined) were significantly different, and the ICG alone is also acceptable, since it has a good performance in sentinel lymph node identification and it can avoid tattooing, with a 100% sentinel lymph node detection rate when combined with patent blue. The higher cost of the ICG dye did not impact the increase in the overall cost of hospitalization.

## Supporting information

**S1 Checklist. TREND statement checklist.**
(DOCX)

**S1 File. Hospital cost of each patient in Brazilian real.**
(DOCX)

**S2 File. Study project in English.**
(DOCX)

**S3 File. Study project in Portuguese.**
(DOCX)

**S1 Data. Data from the 99 patients.**
(XLSX)

**S2 Data. Data English.**
(XLSX)

## Acknowledgments

We are very grateful to the 99 patients participating in the study who made possible the unprecedented research on the indocyanine green dye associated with fluorescence in the breast cancer scenario in Brazil and to the board of Hospital de Esperança for supporting this study. We also would like to thank Patricia Logullo (Palavra Impressa) for language editing services in the last submitted version of this paper.

## Author Contributions

**Conceptualization:** Rafael da Silva Sá, Afonso Celso Pinto Nazário.

**Data curation:** Rafael da Silva Sá.

**Formal analysis:** Rafael da Silva Sá, Suelen Umbelino da Silva, Afonso Celso Pinto Nazário.

**Funding acquisition:** Rafael da Silva Sá, Afonso Celso Pinto Nazário.

**Investigation:** Rafael da Silva Sá, Raquel Fujinohara Von Ah Rodrigues, Luiz Antônio Bugalho.

**Methodology:** Rafael da Silva Sá, Suelen Umbelino da Silva.

**Project administration:** Rafael da Silva Sá, Afonso Celso Pinto Nazário.

**Resources:** Rafael da Silva Sá.

**Software:** Rafael da Silva Sá.

**Supervision:** Rafael da Silva Sá, Afonso Celso Pinto Nazário.

**Validation:** Rafael da Silva Sá, Suelen Umbelino da Silva, Afonso Celso Pinto Nazário.

**Visualization:** Rafael da Silva Sá, Afonso Celso Pinto Nazário.

**Writing – original draft:** Rafael da Silva Sá, Afonso Celso Pinto Nazário.

**Writing – review & editing:** Rafael da Silva Sá, Afonso Celso Pinto Nazário.

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
