## [Decision Letter · Decision Letter 0]

18 Jan 2023

PONE-D-22-22937Evaluation of the efficacy of using indocyanine green associated with fluorescence in sentinel lymph node biopsyPLOS ONE

Dear Dr. Sá,

Thank you for submitting your manuscript to PLOS ONE. After careful consideration, we feel that it has merit but does not fully meet PLOS ONE’s publication criteria as it currently stands. Therefore, we invite you to submit a revised version of the manuscript that addresses the points raised during the review process. The manuscript has been evaluated by two reviewers, and their comments are available below.

The reviewers have raised concerns regarding the reporting, methodology and statistical analysis of this study. 

Could you please revise the manuscript to carefully address the concerns raised?

We look forward to receiving your revised manuscript.

Kind regards,

Johannes Stortz

Staff Editor

PLOS ONE

and https://journals.plos.org/plosone/s/file?id=ba62/PLOSOne_formatting_sample_title_authors_affiliations.pdf.

“The first author (RSS) was supported as a Doctoral Scholar by the National Council for Scientific and Technological Development (CNPq). This study received no other funding.”

“RSS

1

National Council for Scientific and Technological Development (CNPq)

https://www.gov.br/cnpq/pt-br

Reviewers' comments:

Reviewer's Responses to Questions

**Comments to the Author**

1. Is the manuscript technically sound, and do the data support the conclusions?

Reviewer #1: Yes

Reviewer #2: Yes

2. Has the statistical analysis been performed appropriately and rigorously? 

Reviewer #1: Yes

Reviewer #2: Yes

3. Have the authors made all data underlying the findings in their manuscript fully available?

Reviewer #1: Yes

Reviewer #2: Yes

4. Is the manuscript presented in an intelligible fashion and written in standard English?

Reviewer #1: Yes

Reviewer #2: Yes

5. Review Comments to the Author

Reviewer #1: A non-randomized, prospective clinical trial was conducted which aimed to evaluate the efficacy and the sentinel lymph node detection rate when using indocyanine green, an innovative technique. The accuracy rates were 78.8%, 93.9%, and 100% for patent blue, indocyanine green and the combination of the two, respectively.

Minor revisions:

1- Within the abstract and the main section, provide 95% confidence intervals for the detection rates and standard deviations for the mean times of sentinel lymph node detection and surgery.

2- Statistical analysis section: Clarify what is meant by “To compare the values spent . . .”.

3- Before applying Tukey’s multiple comparison tests, check for a significant overall effect. For example, compare the three means using an ANOVA test (overall test), if it is significant then apply Tukey’s multiple comparison tests. If it is not significant, do not apply Tukey’s tests.

4- The standard statistical term for average is mean.

5- Data is missing in table 3 for the combination of dyes.

6- Tukey's test is used for pairwise comparisons. Thus this statement is unclear because the pairs have not been identified. “*According to the p-values of Tukey's multiple comparison test, p=0.013 for the blue-green comparison, p=0.003 for the combined green, and 0.905 for the combined blue.”

7- Figure 2: Add error bars for each group at the two time points.

8- Table 2: Identify the statistical testing method used to estimate each p-value. Since all continuous factors are summarized as means and standard deviations, are all data normally distributed and all p-values from an ANOVA test? If these values are non-normally distributed, summarize using median, first and third quartiles and compare with the Kruskal-Wallis test.

9- To assist in the review process, add line numbering to the document.

Reviewer #2: The manuscript investigated the sentinel lymph node detection rate between the use of patent blue, indocyanine green and patent blue + indocyanine green (combined). Although some similar researches have already published, the specific local studies in Brazil further prove the ICG for the use of sentinel lymph nodes with results better than patent blue. The experimental design was sound, with reasonable data interpretation and result discussion. However, some specific points should be stated clearly as follow：

1. The patients and tumor characteristics should provide as tables in the manuscript.

2. The abbreviation ICG of indocyanine green could used directly in the manuscript except for the first explanation.

3. In section "Materials used", the authors should check the description of "indocyanine green dye 5ml/10ml". It should be “5 mg/10 mL”?

4. Language has to be improved to make the article more attractive.

6. PLOS authors have the option to publish the peer review history of their article (what does this mean?). If published, this will include your full peer review and any attached files.

Reviewer #1: No

Reviewer #2: No

---

## [Author Response · Author response to Decision Letter 0]

27 Feb 2023

Reviewer 1 

Evaluations (peer review comments for the author) 

Reviewer #1: A non-randomized, prospective clinical trial was conducted which aimed to evaluate the efficacy and the sentinel lymph node detection rate when using indocyanine green, an innovative technique. The accuracy rates were 78.8%, 93.9%, and 100% for patent blue, indocyanine green and the combination of the two, respectively.

Minor revisions:

1- Within the abstract and the main section, provide 95% confidence intervals for the detection rates and standard deviations for the mean times of sentinel lymph node detection and surgery.

RESPONSE: Thank you for the observation. Considering that the IC is only valid when both np and nq are higher than ten or npq > 5, we decided not to present the 95% confidence intervals for the detection rates because it didn't occur in our data. Thus, we opted only to show the observed rates, which are still relevant. The standard deviation was presented alongside its mean time, as per request. 

2- Statistical analysis section: Clarify what is meant by “To compare the values spent . . .”.

RESPONSE: We compared the total hospitalization costs of each patient participating in the research, including the dye, to assess the possible socioeconomic impact of using a certain dye.

3- Before applying Tukey’s multiple comparison tests, check for a significant overall effect. For example, compare the three means using an ANOVA test (overall test), if it is significant then apply Tukey’s multiple comparison tests. If it is not significant, do not apply Tukey’s tests.

RESPONSE: The multiple comparison tests occurred as described by the reviewer. The p-values presented in the table refer to the ANOVA, and the subscripted letters identify the groups with significant differences according to Tukey's test. Thus, we first applied the ANOVA, and only when the result was significant, we used Tukey's multiple comparison test. To avoid visually overloading the table, we used the letters to identify the groups with significant statistical differences.

4- The standard statistical term for average is mean.

RESPONSE: This was adjusted in the text and table, thank you.

5- Data is missing in table 3 for the combination of dyes.

RESPONSE: We thank the reviewer for noticing this. This missing data has already been correctly filled in in Table 3.

6- Tukey's test is used for pairwise comparisons. Thus this statement is unclear because the pairs have not been identified. “*According to the p-values of Tukey's multiple comparison test, p=0.013 for the blue-green comparison, p=0.003 for the combined green, and 0.905 for the combined blue.”

RESPONSE: We adjusted table 3 to keep the same p-value presentation pattern of table 2. The p-value refers to the ANOVA test, and the letters identify statistically different groups, as per Tukey's test. There were no significant differences between the combined and blue dyes, so we omitted the letter A. 

7- Figure 2: Add error bars for each group at the two time points.

RESPONSE: We did not add error bars to the chart for the same reason cited in the item's one response, considering that error bars usually represent standard error (SE). 

8- Table 2: Identify the statistical testing method used to estimate each p-value. Since all continuous factors are summarized as means and standard deviations, are all data normally distributed and all p-values from an ANOVA test? If these values are non-normally distributed, summarize using median, first and third quartiles and compare with the Kruskal-Wallis test.

RESPONSE: We adjusted table 2 to clarify the method of p-value estimation. The only variable for which we used the Kruskall-Wallis test was the number of identified lymph nodes (to compare the mean group). When the dye comparison was presented using Qui-Square, we removed the line comparing the means. Thus, the ANOVA test compared all quantitative variables between the groups, while multiple comparisons were through Tukey's test. Before applying the ANOVA test, we tested the normality of the data through the Shapiro-Wilk test. 

9- To assist in the review process, add line numbering to the document.

RESPONSE: We added line numbering in the manuscript to assist in the review process; thank you for this suggestion.

Reviewer 2 

Evaluations (peer review comments for the author) 

Reviewer #2: The manuscript investigated the sentinel lymph node detection rate between the use of patent blue, indocyanine green and patent blue + indocyanine green (combined). Although some similar researches have already published, the specific local studies in Brazil further prove the ICG for the use of sentinel lymph nodes with results better than patent blue. The experimental design was sound, with reasonable data interpretation and result discussion. However, some specific points should be stated clearly as follow：

The patients and tumor characteristics should provide as tables in the manuscript.

RESPONSE: Thank you for the observation. We added the information to the two new tables (Table 1 and Table 3). 

2. The abbreviation ICG of indocyanine green could used directly in the manuscript except for the first explanation.

RESPONSE: Thank you for the suggestion. We updated the manuscript with the abbreviation.

3. In section "Materials used", the authors should check the description of "indocyanine green dye 5ml/10ml". It should be “5 mg/10 mL”?

RESPONSE: Exactly. We apologize for this error and have already corrected this unit in the text.

4. Language has to be improved to make the article more attractive.

RESPONSE: We had the language reviewed. We thank the reviewer for the collaboration with our manuscript.

Editorial Corrections (Journal requirements):

1. Please ensure that your manuscript meets PLOS ONE's style requirements, including those for file naming. The PLOS ONE style templates can be found at https://journals.plos.org/plosone/s/file?id=wjVg/PLOSOne_formatting_sample_main_body.pdf and https://journals.plos.org/plosone/s/file?id=ba62/PLOSOne_formatting_sample_title_authors_affiliations.pdf.

RESPONSE: We edited and improved the manuscript with the PLOS ONE's style requirements.

2. Thank you for stating the following in the Acknowledgments Section of your manuscript: “The first author (RSS) was supported as a Doctoral Scholar by the National Council for Scientific and Technological Development (CNPq). This study received no other funding.” We note that you have provided funding information that is not currently declared in your Funding Statement. However, funding information should not appear in the Acknowledgments section or other areas of your manuscript. We will only publish funding information present in the Funding Statement section of the online submission form. Please remove any funding-related text from the manuscript and let us know how you would like to update your Funding Statement. Currently, your Funding Statement reads as follows: “RSS 1 National Council for Scientific and Technological Development (CNPq) https://www.gov.br/cnpq/pt-br. The funders had no role in study design, data collection and analysis, decision to publish, or preparation of the manuscript.” Please include your amended statements within your cover letter; we will change the online submission form on your behalf.

RESPONSE: We corrected the acknowledgments section.

3. We note that you have indicated that data from this study are available upon request. PLOS only allows data to be available upon request if there are legal or ethical restrictions on sharing data publicly. For information on unacceptable data access restrictions, please see http://journals.plos.org/plosone/s/data-availability#loc-unacceptable-data-access-restrictions. In your revised cover letter, please address the following prompts: a) If there are ethical or legal restrictions on sharing a de-identified data set, please explain them in detail (e.g., data contain potentially identifying or sensitive patient information) and who has imposed them (e.g., an ethics committee). Please also provide contact information for a data access committee, ethics committee, or other institutional body to which data requests may be sent. b) If there are no restrictions, please upload the minimal anonymized data set necessary to replicate your study findings as either Supporting Information files or to a stable, public repository and provide us with the relevant URLs, DOIs, or accession numbers. Please see http://www.bmj.com/content/340/bmj.c181.long for guidelines on how to de-identify and prepare clinical data for publication. For a list of acceptable repositories, please see http://journals.plos.org/plosone/s/data-availability#loc-recommended-repositories. We will update your Data Availability statement on your behalf to reflect the information you provide. 

RESPONSE: We are providing the anonymized data upload set necessary to replicate your study findings. This study does not have an ethical or legal restriction on sharing an unidentified dataset. You can find the data attached in a separate file.

---

## [Decision Letter · Decision Letter 1]

17 Apr 2023

PONE-D-22-22937R1Evaluation of the efficacy of using indocyanine green associated with fluorescence in sentinel lymph node biopsyPLOS ONE

Dear Dr. Sá,

Thank you for submitting your manuscript to PLOS ONE. After careful consideration, we feel that it has merit but does not fully meet PLOS ONE’s publication criteria as it currently stands. Therefore, we invite you to submit a revised version of the manuscript that addresses the points raised during the review process. Your manuscript has been evaluated by two reviewers; one from the previous round of review, and one new reviewer. While both reviewers are positive regarding the study, concerns have been raised regarding the language quality and overall scientific readability of your manuscript, not limited to the examples given in the comments. Please ensure you thoroughly copyedit your manuscript; we strongly recommend that you utilize a fluent English speaker or a professional editing service. When you resubmit the manuscript, please update your cover letter to include the name of the colleague or professional editing service that assisted you.

We look forward to receiving your revised manuscript.

Kind regards,

Hugh Cowley

Staff Editor

PLOS ONE

Reviewers' comments:

Reviewer's Responses to Questions

**Comments to the Author**

1. If the authors have adequately addressed your comments raised in a previous round of review and you feel that this manuscript is now acceptable for publication, you may indicate that here to bypass the “Comments to the Author” section, enter your conflict of interest statement in the “Confidential to Editor” section, and submit your "Accept" recommendation.

Reviewer #1: All comments have been addressed

Reviewer #3: (No Response)

2. Is the manuscript technically sound, and do the data support the conclusions?

Reviewer #1: (No Response)

Reviewer #3: Partly

3. Has the statistical analysis been performed appropriately and rigorously? 

Reviewer #1: (No Response)

Reviewer #3: Yes

4. Have the authors made all data underlying the findings in their manuscript fully available?

Reviewer #1: (No Response)

Reviewer #3: Yes

5. Is the manuscript presented in an intelligible fashion and written in standard English?

Reviewer #1: (No Response)

Reviewer #3: No

6. Review Comments to the Author

Reviewer #1: (No Response)

Reviewer #3: This study compared 3 different methods to detect the axillary sentinel lymph nodes in breast cancer surgery; blue dye alone, ICG fluorescence alone, and their combination. They reported the significant improvement in the detection rate and procedure time of sentinel node biopsy when ICG fluorescence was used. Although the study was relatively small, with 33 cases in each group, and was a serial and retrospective comparison, the findings were very encouraging considering it was the first study to demonstrated the feasibility of ICG fluorescent method in Brazilian community. I sincerely respect the authors for their massive efforts and enthusiasm, and totally agreed that this kind of data could be important basis for the implementation of ICG fluorescence into Brazilian society. However, I found a substantial number of grammatical errors in the manuscript. In addition, the style of the manuscript was not organized well as a scientific manuscript. Thus, in a lot of parts of the manuscript, I could not follow what the authors really wanted to address. The following are examples which should to be corrected. Collectively, I found the manuscript was basically encouraging but not to be accepted in its current form. I strongly recommend the authors have it professionally edited and submit it as a separate manuscript.

- Page6 line 4: “patent blue dye v 50mg/2ml”. “v” does not make sense.

- Page6 line7: “5ml/10ml” might be correctly “5mg/10ml”

- Page6 line8; “periareolar infltration” should be “periareolar injection”

- Page6 line 10-11: I think this sentence contains multiple grammatical errors.

- Page6 line 19: The term “axillary dissection” is misleading since it normally means axillary cleanance.

- Page6 line22: “10ml/5mg” should be “5mg/10ml”.

- Page7 line6: Although I can follow what the author wants to say here, the sentence looks wired.

- Page8 line 3: the sentence doesn’t make sense.

- Page8 line7 to 16: this part doesn’t make sense.

- Page8 line21: “Time of identification” should be “Time to identification”.

- Page10 line6; ”menacme” might be “menarche”.

- From page 11: please separate the tables and figures from the body of manuscript.

- Page 11 to 12: the contents written in S2 Table should not be addressed repeatedly in the body of manuscripts.

- Page12 line 10: Please provide the definition of “accuracy rate”.

- Page14 line 7: “amounts spent” does not make sense.

- Page14 line 10: please spell out “BRL” when it firstly appears in the manuscript.

- Page15 1st section of the discussion: I agreed with the authors’ statement.

7. PLOS authors have the option to publish the peer review history of their article (what does this mean?). If published, this will include your full peer review and any attached files.

Reviewer #1: No

Reviewer #3: No

---

## [Author Response · Author response to Decision Letter 1]

7 May 2023

Dear Hugh Cowley

Thank you for your consideration of our paper for publication in PLOS ONE, and also for the positive reviews. 

We specially thank Reviewer #3 for the careful review of the manuscript, and we would like them to know that we had the text professionally edited, with all the corrections listed made and also further improvements. We believe the paper is now ready for publication.

Please let us know if you have any other questions.

Thank you very much

Rafael S. Sá and co-authors

Reviewers' comments: 

Reviewer's Responses to Questions

Comments to the Author  1. If the authors have adequately addressed your comments raised in a previous round of review and you feel that this manuscript is now acceptable for publication, you may indicate that here to bypass the “Comments to the Author” section, enter your conflict of interest statement in the “Confidential to Editor” section, and submit your "Accept" recommendation.

Reviewer #1: All comments have been addressed

Reviewer #3: (No Response)

RESPONSE: Thank you for the positive evaluation. We have corrected all issues raised in the last review, and went further with a text editing and more corrections.

2. Is the manuscript technically sound, and do the data support the conclusions?  The manuscript must describe a technically sound piece of scientific research with data that supports the conclusions. Experiments must have been conducted rigorously, with appropriate controls, replication, and sample sizes. The conclusions must be drawn appropriately based on the data presented. 

Reviewer #1: (No Response)

Reviewer #3: Partly

RESPONSE: As mentioned above, we have addressed all issues raised in the last review, and went further with a text editing and more corrections. This made the text clearer and we are sure that Reviewer #3 can now be confident about the robustness of our research methods and reporting.

3. Has the statistical analysis been performed appropriately and rigorously? 

Reviewer #1: (No Response)

Reviewer #3: Yes

4. Have the authors made all data underlying the findings in their manuscript fully available?  The PLOS Data policy requires authors to make all data underlying the findings described in their manuscript fully available without restriction, with rare exception (please refer to the Data Availability Statement in the manuscript PDF file). The data should be provided as part of the manuscript or its supporting information, or deposited to a public repository. For example, in addition to summary statistics, the data points behind means, medians and variance measures should be available. If there are restrictions on publicly sharing data—e.g. participant privacy or use of data from a third party—those must be specified.

Reviewer #1: (No Response)

Reviewer #3: Yes

RESPONSE: Thank you for the positive reviews in items 3 and 4 above.

5. Is the manuscript presented in an intelligible fashion and written in standard English?  PLOS ONE does not copyedit accepted manuscripts, so the language in submitted articles must be clear, correct, and unambiguous. Any typographical or grammatical errors should be corrected at revision, so please note any specific errors here.

Reviewer #1: (No Response)

Reviewer #3: No

RESPONSE: As mentioned above, we have hired professional text editing and made more corrections to the text. This made the text content clearer and the reading flows better now.

6. Review Comments to the Author  Please use the space provided to explain your answers to the questions above. You may also include additional comments for the author, including concerns about dual publication, research ethics, or publication ethics. (Please upload your review as an attachment if it exceeds 20,000 characters)

Reviewer #1: (No Response)

Reviewer #3: This study compared 3 different methods to detect the axillary sentinel lymph nodes in breast cancer surgery; blue dye alone, ICG fluorescence alone, and their combination. They reported the significant improvement in the detection rate and procedure time of sentinel node biopsy when ICG fluorescence was used. Although the study was relatively small, with 33 cases in each group, and was a serial and retrospective comparison, the findings were very encouraging considering it was the first study to demonstrated the feasibility of ICG fluorescent method in Brazilian community. I sincerely respect the authors for their massive efforts and enthusiasm, and totally agreed that this kind of data could be important basis for the implementation of ICG fluorescence into Brazilian society. However, I found a substantial number of grammatical errors in the manuscript. In addition, the style of the manuscript was not organized well as a scientific manuscript. Thus, in a lot of parts of the manuscript, I could not follow what the authors really wanted to address. The following are examples which should to be corrected. Collectively, I found the manuscript was basically encouraging but not to be accepted in its current form. I strongly recommend the authors have it professionally edited and submit it as a separate manuscript.  - Page6 line 4: “patent blue dye v 50mg/2ml”. “v” does not make sense.

- Page6 line7: “5ml/10ml” might be correctly “5mg/10ml” - Page6 line8; “periareolar infltration” should be “periareolar injection” - Page6 line 10-11: I think this sentence contains multiple grammatical errors. - Page6 line 19: The term “axillary dissection” is misleading since it normally means axillary cleanance. - Page6 line22: “10ml/5mg” should be “5mg/10ml”. - Page7 line6: Although I can follow what the author wants to say here, the sentence looks wired. - Page8 line 3: the sentence doesn’t make sense. - Page8 line7 to 16: this part doesn’t make sense. - Page8 line21: “Time of identification” should be “Time to identification”. - Page10 line6; ”menacme” might be “menarche”. - From page 11: please separate the tables and figures from the body of manuscript. - Page 11 to 12: the contents written in S2 Table should not be addressed repeatedly in the body of manuscripts. - Page12 line 10: Please provide the definition of “accuracy rate”. - Page14 line 7: “amounts spent” does not make sense. - Page14 line 10: please spell out “BRL” when it firstly appears in the manuscript. - Page15 1st section of the discussion: I agreed with the authors’ statement.

RESPONSE: During the professional editing we hired for this manuscript, all items above have been corrected and the info added to the manuscript, except for thre issues: 

1) the word menacme, which is correct in the context of our study. Menarche is defined as the first menstrual period in a woman’s life, which does not apply here. In our study, almost 30% of women were in the menacme phase, meaning “the period of a woman’s life cycle in which she experiences menstrual activity”. Reference for this definition here: https://dictionary.apa.org/menacme. 

2) we do not understand what the reviewer means by “S2 Table” — we believe that all tables should be cited in the text, and this is what we tried to do.

3) We did not remove the tables from the manuscript document, as this is required in the journal’s Instructions for Authors: “Place each table in your manuscript file directly after the paragraph in which it is first cited (read order)”.

7. PLOS authors have the option to publish the peer review history of their article (what does this mean?). If published, this will include your full peer review and any attached files. Do you want your identity to be public for this peer review? For information about this choice, including consent withdrawal, please see our Privacy Policy.

Reviewer #1: No

Reviewer #3: No

---

## [Decision Letter · Decision Letter 2]

13 Jun 2023

PONE-D-22-22937R2Evaluation of the efficacy of using indocyanine green associated with fluorescence in sentinel lymph node biopsyPLOS ONE

Dear Dr. Rafael da Silva Sá,

Thank you for submitting your manuscript to PLOS ONE. After careful consideration, we feel that it has merit but does not fully meet PLOS ONE’s publication criteria as it currently stands. Therefore, we invite you to submit a revised version of the manuscript that addresses the points raised during the review process. Please find below additional revision requests made by the reviewers.  Please submit your revised manuscript by Jul 28 2023 11:59PM. If you will need more time than this to complete your revisions, please reply to this message or contact the journal office at plosone@plos.org. Please include the following items when submitting your revised manuscript:A rebuttal letter that responds to each point raised by the academic editor and reviewer(s). You should upload this letter as a separate file labeled 'Response to Reviewers'.A marked-up copy of your manuscript that highlights changes made to the original version. You should upload this as a separate file labeled 'Revised Manuscript with Track Changes'.An unmarked version of your revised paper without tracked changes. You should upload this as a separate file labeled 'Manuscript'.If applicable, we recommend that you deposit your laboratory protocols in protocols.io to enhance the reproducibility of your results. Protocols.io assigns your protocol its own identifier (DOI) so that it can be cited independently in the future. For instructions see: https://journals.plos.org/plosone/s/submission-guidelines#loc-laboratory-protocols. Additionally, PLOS ONE offers an option for publishing peer-reviewed Lab Protocol articles, which describe protocols hosted on protocols.io. Read more information on sharing protocols at https://plos.org/protocols?utm_medium=editorial-email&utm_source=authorletters&utm_campaign=protocols.

We look forward to receiving your revised manuscript.

Kind regards,

Salih Colakoglu

Academic Editor

PLOS ONE

Journal Requirements:

Reviewers' comments:

Reviewer's Responses to Questions

**Comments to the Author**

1. If the authors have adequately addressed your comments raised in a previous round of review and you feel that this manuscript is now acceptable for publication, you may indicate that here to bypass the “Comments to the Author” section, enter your conflict of interest statement in the “Confidential to Editor” section, and submit your "Accept" recommendation.

Reviewer #1: (No Response)

Reviewer #3: (No Response)

2. Is the manuscript technically sound, and do the data support the conclusions?

Reviewer #1: Yes

Reviewer #3: Partly

3. Has the statistical analysis been performed appropriately and rigorously? 

Reviewer #1: Yes

Reviewer #3: Yes

4. Have the authors made all data underlying the findings in their manuscript fully available?

Reviewer #1: Yes

Reviewer #3: Yes

5. Is the manuscript presented in an intelligible fashion and written in standard English?

Reviewer #1: Yes

Reviewer #3: Yes

6. Review Comments to the Author

Reviewer #1: Minor revisions:

1- Line 48: After "mean" add "+/- SD (standard deviation)."

2- Line 52: Indicate the statistical testing method used to estimate p<0.001.

3- Line 207: For clarity, simply state, "The data was collected in a Microsoft Excel spreadsheet prepared by the statistician."

4- Line 308: Clarify: "normally distributed variables."

5- Line 309: Clarify: "non-normally distributed variables."

6- Line 315: Consider replacing the following statement: "The significance level adopted in all tests was 5%." with "P-values less than 0.05 were considered statistically significant."

7- The phrase "no report of male patients" in line 330 is awkward.

8- Tables 1 and 3: Distinguish the row for age from those for menopausal status.

9- Table 3: Merge the cells in the "BMI" header row to make it consistent with the rest of the table.

10- Line 436: The standard statistical term for average is mean.

11- Figure 2: Label the y-axis: "Percentage".

12- Figure 3: Label the y-axis.

Note: Line numbers refer to those in the tracked changes version of revision 2.

Reviewer #3: Thanks for your re-submission. English has been improved and now I can review the manuscript in depth. I think the manuscript is still to be revised.

- First of all, I have some comments on Figures.

The unnecessary subtitle and the logo of Fig1 (CONSORT TRNAPARENT,,, and Evaluation of the efficacy,,,) should be removed.

- In Fig 4, Circle chart is not appropriate to present this kind of data. Use bar chart or Table. I think the information of Fig2 and 4 should be presented by using Table. These may be able to incorporated into Table 4 or the new table with similar structure.

- Title of Fig 3 should be considered again, I suppose that many readers cannot understand what is the "rate" of sentinel lymph node identification time. And consider the style of the graph again. Dot plot with connecting bars does not fit for this kind of data. Use table if you cannot determine the proper style.

- Line 168: please correct the spacing of “5 mg/10 ml”

- Line 179, Line 185, and many others throughout the manuscript: As I mentioned before, please use "injection" rather than "infiltration". Most of the article use the term "injection".

- Line 261: Similarly, the word "menacme" is not commonly used. Please using the term "premenopausal" is much better.

- Line 182 and 189: As I mentioned before, since you did not perform ”axillary clearance” in many cases, you need to rephrase the term to avoid the reader’s confusion. For instance, I suggest the use the word “axillary surgery”.

- Line 355-367: There are a lot of trials evaluating the detection rate of ICG fluorescence in comparison with blue dye and its usefulness has been widely accepted although many clinical guidelines do not mention it. Therefore, the authors should cite at least the following meta-analysis and improve their discussion. In my understanding, this study is a kind of validation study of ICG fluorescence in Brazil.

1. Lancet Oncol. 2014 Jul;15(8):e351-62. doi: 10.1016/S1470-2045(13)70590-4.

2. PLoS One. 2016 Jun 9;11(6):e0155597. doi: 10.1371/journal.pone.0155597. eCollection 2016.

3. Eur J Surg Oncol. 2014 Jul;40(7):843-9. doi: 10.1016/j.ejso.2014.02.228. Epub 2014 Feb 25.

- Line 365-367: Can the author compare the BMI or other factors between the patients whose sentinel node could be identified and those could not be identified?

- Line399-409: Why don’t you discuss about the concentration of ICG more precisely? You will be able to easily find that most of the trials use ICG at the concentration of 2.5mg/ml to 5mg/ml and injection dose is 0.5 ml to 1.0ml and reported good detection rate in SLNB. This issue has already discussed in a textbook; Yamagami, K, et al. Practice of Fluorescence Navigation Surgery Using Indocyanine Green for Sentinel Lymph Node Biopsy in Breast Cancer. In: Kusano, M., Kokudo, N., Toi, M., Kaibori, M. (eds) ICG Fluorescence Imaging and Navigation Surgery. Springer, Tokyo. 2016, pp 113–123. If the authors think this study is a kind of landmark trial in Brazil, such practical information should be provided here.

- Line 451: In terms of long-term follow up, these two articles should be the largest ones rather than the report by Shen et al.

1. Breast Cancer Res Treat. 2021 Jul;188(2):361-368. doi: 10.1007/s10549-021-06196-6. Epub 2021 Mar 24

2. Int J Clin Oncol. 2021 Aug;26(8):1461-1468. doi: 10.1007/s10147-021-01925-9. Epub 2021 Apr 20

- Line 300: Again, what is “accuracy rate”? Please give us the definition. It means detection rate?

- Line 390-398: I think this part of the discussion is not relevant to this study. Is it necessary?

- Can authors provide the concordance rate between blue dye and ICG in the combination arm? In other word, how many nodes can be detected only by ICG or vice versa.

- Please discuss on the result of number of identified lymph node. It should be a key finding of this study but I could not find the discussion on the date in the manuscript.

- Conclusions: “The combined method was the most effective” is overstated. Your data shows ICG alone is also acceptable since it has a good performance in SLN identification and it can avoid tattooing.

7. PLOS authors have the option to publish the peer review history of their article (what does this mean?). If published, this will include your full peer review and any attached files.

Reviewer #1: No

Reviewer #3: No

---

## [Author Response · Author response to Decision Letter 2]

10 Jul 2023

Reviewers’ questions 1-5 and 7 don’t need authors’ responses.

6. Review Comments to the Author  

Reviewer #1: Minor revisions: 1- Line 48: After "mean" add "+/- SD (standard deviation)." – CORRECTED, THANK YOU. 2- Line 52: Indicate the statistical testing method used to estimate p<0.001. – INFORMATION ADDED, THANK YOU.

3- Line 207: For clarity, simply state, "The data was collected in a Microsoft Excel spreadsheet prepared by the statistician." – CORRECTED, THANK YOU. 4- Line 308: Clarify: "normally distributed variables." – CORRECTED, THANK YOU. 5- Line 309: Clarify: "non-normally distributed variables." – CORRECTED, THANK YOU. 6- Line 315: Consider replacing the following statement: "The significance level adopted in all tests was 5%." with "P-values less than 0.05 were considered statistically significant." – MODIFIED ACCORDING TO REVIEWER’S TASTE. 7- The phrase "no report of male patients" in line 330 is awkward. – CORRECTED, THANK YOU. 8- Tables 1 and 3: Distinguish the row for age from those for menopausal status. – CORRECTED, THANK YOU. 9- Table 3: Merge the cells in the "BMI" header row to make it consistent with the rest of the table. – CORRECTED, THANK YOU. 10- Line 436: The standard statistical term for average is mean. – MODIFIED ACCORDING TO REVIEWER’S TASTE. 11- Figure 2: Label the y-axis: "Percentage".– CORRECTED, THANK YOU. 12- Figure 3: Label the y-axis. – CORRECTED, THANK YOU. Note: Line numbers refer to those in the tracked changes version of revision 2.

Reviewer #3: Thanks for your re-submission. English has been improved and now I can review the manuscript in depth. I think the manuscript is still to be revised.

- First of all, I have some comments on Figures. The unnecessary subtitle and the logo of Fig1 (CONSORT TRNAPARENT,,, and Evaluation of the efficacy,,,) should be removed. – CORRECTED, THANK YOU. - In Fig 4, Circle chart is not appropriate to present this kind of data. Use bar chart or Table. I think the information of Fig2 and 4 should be presented by using Table. These may be able to incorporated into Table 4 or the new table with similar structure. – THE DATA IS ALREADY PRESENTED IN TABLE 4, SO WE DELETED THE FIGURE.

- Title of Fig 3 should be considered again, I suppose that many readers cannot understand what is the "rate" of sentinel lymph node identification time. And consider the style of the graph again. Dot plot with connecting bars does not fit for this kind of data. Use table if you cannot determine the proper style. – THANK YOU FOR THE SUGGESTION. WE CREATED A BAR CHART AND ADDED IT TO THE MANUSCRIPT WITH A NEW TITLE.

- Line 168: please correct the spacing of “5 mg/10 ml” – SPACING HAS BEEN STANDARDIZED FOR ALL, THANK YOU. - Line 179, Line 185, and many others throughout the manuscript: As I mentioned before, please use "injection" rather than "infiltration". Most of the article use the term "injection". – CORRECTED, THANK YOU. - Line 261: Similarly, the word "menacme" is not commonly used. Please using the term "premenopausal" is much better. – MODIFIED ACCORDING TO REVIEWER’S PREFERENCE. - Line 182 and 189: As I mentioned before, since you did not perform ”axillary clearance” in many cases, you need to rephrase the term to avoid the reader’s confusion. For instance, I suggest the use the word “axillary surgery”. 0 MODIFIED.

- Line 355-367: There are a lot of trials evaluating the detection rate of ICG fluorescence in comparison with blue dye and its usefulness has been widely accepted although many clinical guidelines do not mention it. Therefore, the authors should cite at least the following meta-analysis and improve their discussion. In my understanding, this study is a kind of validation study of ICG fluorescence in Brazil. – THANK YOU FOR THE SUGGESTION, WE ADDED TEXT CITING TWO OF THESE STUDIES. 1. Lancet Oncol. 2014 Jul;15(8):e351-62. doi: 10.1016/S1470-2045(13)70590-4. 2. PLoS One. 2016 Jun 9;11(6):e0155597. doi: 10.1371/journal.pone.0155597. eCollection 2016. 3. Eur J Surg Oncol. 2014 Jul;40(7):843-9. doi: 10.1016/j.ejso.2014.02.228. Epub 2014 Feb 25.

- Line 365-367: Can the author compare the BMI or other factors between the patients whose sentinel node could be identified and those could not be identified? – WE DID COMPARE AND FOUND NO SIGNIFICANT DIFFERENCE. THIS ANALYSIS WAS ADDED TO THE TEXT.

- Line399-409: Why don’t you discuss about the concentration of ICG more precisely? You will be able to easily find that most of the trials use ICG at the concentration of 2.5mg/ml to 5mg/ml and injection dose is 0.5 ml to 1.0ml and reported good detection rate in SLNB. This issue has already discussed in a textbook; Yamagami, K, et al. Practice of Fluorescence Navigation Surgery Using Indocyanine Green for Sentinel Lymph Node Biopsy in Breast Cancer. In: Kusano, M., Kokudo, N., Toi, M., Kaibori, M. (eds) ICG Fluorescence Imaging and Navigation Surgery. Springer, Tokyo. 2016, pp 113–123. If the authors think this study is a kind of landmark trial in Brazil, such practical information should be provided here. – WE THANK THE REVIEWER FOR THE SUGGESTION. WE READ THE STUDY AND DISCUSSED ITS FINDINGS IN THE MANUSCRIPT.

- Line 451: In terms of long-term follow up, these two articles should be the largest ones rather than the report by Shen et al. 1. Breast Cancer Res Treat. 2021 Jul;188(2):361-368. doi: 10.1007/s10549-021-06196-6. Epub 2021 Mar 24 2. Int J Clin Oncol. 2021 Aug;26(8):1461-1468. doi: 10.1007/s10147-021-01925-9. Epub 2021 Apr 20 – MODIFIED; STUDIES CITED, THANK YOU. - Line 300: Again, what is “accuracy rate”? Please give us the definition. It means detection rate? – CORRECTED TO “DETECTION RATE”, THANK YOU.

- Line 390-398: I think this part of the discussion is not relevant to this study. Is it necessary? – WE DELETED THE PARAGRAPH.

- Can authors provide the concordance rate between blue dye and ICG in the combination arm? In other word, how many nodes can be detected only by ICG or vice versa. – THANK YOU FOR THE SUGGESTION, TABLE ADDED (AS TABLE 6).

- Please discuss on the result of number of identified lymph node. It should be a key finding of this study but I could not find the discussion on the date in the manuscript. – THANK YOU FOR THE SUGGESTION, TEXT ADDED TO THE DISCUSSION. - Conclusions: “The combined method was the most effective” is overstated. Your data shows ICG alone is also acceptable since it has a good performance in SLN identification and it can avoid tattooing. – CORRECTED, THANK YOU.

---

## [Editor Report · Decision Letter 3]

9 Aug 2023

Evaluation of the efficacy of using indocyanine green associated with fluorescence in sentinel lymph node biopsy

PONE-D-22-22937R3

Dear Dr. Rafael da Silva Sa,

We’re pleased to inform you that your manuscript has been judged scientifically suitable for publication and will be formally accepted for publication once it meets all outstanding technical requirements.

Kind regards,

Salih Colakoglu

Academic Editor

PLOS ONE

---

## [Editor Report · Acceptance letter]

15 Aug 2023

PONE-D-22-22937R3 

Evaluation of the efficacy of using indocyanine green associated with fluorescence in sentinel lymph node biopsy 

Dear Dr. Sá:

I'm pleased to inform you that your manuscript has been deemed suitable for publication in PLOS ONE. Congratulations! Your manuscript is now with our production department. 

Kind regards, 

on behalf of

Dr. Salih Colakoglu 

Academic Editor

PLOS ONE